# Progressive Dynamic Registration Method for Tile Maps Based on Optimum Multi-Features

**Dong Zhang and Jiqiu Deng ***

School of Geosciences and Info-Physics, Central South University, Changsha 410083, China
* Correspondence: csugis@csu.edu.cn

**Abstract:** Different tile maps may use different coordinate systems, and it is difficult to superimpose maps from different sources. In this regard, we propose a progressive dynamic registration (PDR) method based on optimum features extracted from images of tile map screenshots of a certain scale. Using this method, we can automatically register maps in roughly the same area from different sources without knowing the map project information. The better features among feature points and feature surfaces are selected for image registration based on the richness of the features in the map, and a new matching filter that combines the characteristics of the tile map and the features is proposed. In the progressive registration process during map zooming, the result of the adjacent scales are used as reference values for coarse registration. After experimental verification in different areas, the root mean square error of PDR is below 2.5 in different maps and is better than that of the registration method using feature points only. Moreover, the registration accuracies of remote sensing maps in all areas and vector maps in nonurban areas are better than that of the method based on coordinate system transformation. The calculation results of PDR can register not only the tile maps but also other nontiled vector or remote sensing data.

**Keywords:** tile map; map registration; feature points; feature surfaces; feature matching

## 1. Introduction

The most widely used coordinate system for online tile maps is the WGS-84 coordinate system. Google Maps, Bing Maps, and Open Street Map usually use WGS-84, while some maps use other coordinate systems. For example, Tianditu uses CGCS2000, Google Maps and Amap within China use GCJ-02, which was introduced by the National Bureau of Surveying and Mapping of China, and Baidu Maps uses BD-09. Some niche maps may use local independent coordinate systems. There are differences in display results between different coordinate systems [1], so data from different sources will be offset when overlapped and used [2]. Therefore, before using maps from different sources simultaneously, it is necessary to register maps with different coordinate systems. Map registration refers to the process of configuring a map with an unknown or different coordinate system to a map with a known coordinate system. The traditional method based on the formula needs to include the coordinate projection formula of the two maps, and the usage scenarios are relatively limited. The implementation process of semi-automatic map registration involves selecting some corresponding points between different maps as control points by computer or human, calculating the registration parameters, and then using the registration tool to perform projection transformation on the map to be registered [3]. Among the fully automatic registration methods, the feature-based registration method can automatically acquire image features for registration, and it is currently the most widely used method of image automatic registration [4].

Using the registration method based on image features, we only need to obtain the map image service to complete the registration, and the registration result is more accurate to a certain extent. Different feature elements are used to match different types of maps

and different regions. Compared with the registration method that only uses feature points, the PDR method has better matching accuracy and wider application scenarios.

Feature-based automatic image registration usually includes four steps: (1) feature extraction; (2) feature matching; (3) determining the image transformation model; (4) image transformation and interpolation [5]. In the feature extraction stage, the features can be divided into feature points, feature lines, and feature surfaces [6].

## 2. Literature Review

In terms of feature points, Moravec [7] proposed the Moravec corner point detection algorithm in 1977. He used a sliding window with a W × W range and determined whether the current pixel point is an edge or a corner point based on the pixel change E when the window is moved in different directions. The algorithm calculates the change of E in only four main directions. Therefore, the algorithm is fast, but is sensitive to noise and edges and has directional limitations. Due to the shortcomings of Moravec's algorithm, in 1988, Harris [8] proposed the Harris operator which computes the pixel change when the window is moved in any direction. He used Taylor expansion to approximate the arbitrary orientation and determined whether a pixel point within the window is a corner point based on the two eigenvalues of the binomial function. Lowe proposed a feature extraction algorithm which is based on scale-invariant feature transform (SIFT) in 1999 [9] and improved it in 2004 [10]. The SIFT algorithm simulates images of different scales by constructing image pyramids, identifying the extreme points in the pyramid as candidate feature points, then removing edge effects and discarding some low-contrast points. To ensure the orientation invariance of feature points, each feature point has an orientation. Combining the orientation information of the neighboring sub-blocks of the current point forms a 128-dimensional descriptor. SIFT feature points have good stability and invariance to rotation, scale, and luminance changes, but the extraction speed is slow. In 2006, Bay [11] proposed the speed-up robust features (SURF) algorithm, which improves the convolution speed with a box filter and becomes three times more efficient compared to the SIFT algorithm. Its principle will be described in detail below. In 2011, E Rublee et al. [12] proposed the Oriented FAST and Rotated BRIEF (ORB) feature detection operator, which uses the FAST feature detection and BRIEF descriptors. The speed of the ORB algorithm is much faster than the SIFT algorithm and the SURF algorithm. Since then, many studies that use these feature points or improve on them to solve problems have been conducted, such as image stitching, image fusion, and remote sensing image registration. In 2019, Liu et al. [13] used the SURF algorithm to extract image feature points and combined it with the improved RANSAC algorithm for matching screening to achieve fast stitching of a large number of images. Aiming at low accuracy of feature points matching, Wang et al. [14] proposed an image-matching algorithm that is based on SURF and approximate nearest neighbor search. Fast-Hessian was used to detect the feature points and then SURF feature descriptors were generated. He used RANSAC to eliminate the mismatching points. The proposed matching algorithm improved the matching accuracy rate and real-time performance. In 2022, Qiu et al. [15] proposed the AT-SIFT algorithm with adaptive thresholding for image registration based on the SIFT algorithm to solve the problems of large computation and low matching accuracy, and used the adaptive thresholding FLANN algorithm and the improved RANSAC algorithm for matching filtering, which led to a large improvement in registration accuracy and efficiency.

In terms of feature lines and feature surfaces, in 1968, Irwin Sobel proposed the Sobel operator in a workshop, and the algorithm differentiates or second-order differentiates each pixel of the image according to the different reasons for edge formation, so that the points with significant changes can be detected. Its extraction speed is fast, but the extracted edges are often discontinuous. The Laplace operator is a second-order differential linear operator with stronger edge localization capability. Since the operator is very sensitive to image noise, it is often used by first smoothing the image with a Gaussian smoothing filter, a method called Laplacian of Gaussian (LoG). The Canny edge detection algorithm

was proposed by John F. Canny [16] in 1986 and is still widely used today. The key to the Canny algorithm is to find the location in the image where the gray intensity changes most strongly. The algorithm first selects a certain Gaussian filter for smooth filtering to realize image denoising and then conducts nonpolar suppression and double-threshold screening to obtain the result. There are some other edge detection algorithms, such as ratio of exponential weighted average (ROEWA) [17], line segment detector (LSD) [18], and Gaussian-Gamma-shaped (GGS) bi-window [19], which are also widely used to extract edges from remote sensing images. Gan et al. [20] proposed a registration method based on image feature lines in 2010, which first extracts image edges using the LoG operator, then obtains the main straight lines in the edge information using Hough transform, obtains the registration parameters by calculation, and finally realizes image registration. In 2011, Gao et al. [21] used the Canny algorithm to extract the image edges to obtain the chain code in the image to solve the problem of not matching images under different wavelength channels in NIR ice detection. By comparing the angle and coordinate difference of the chain code, the image was rotated and panned to achieve image matching. In 2017, Li et al. [22] proposed a line segment-based image registration method. He used an improved Canny operator (Canny DIR) to detect edges, then extracted line segments from the edges. The algorithm was reliable and accurate on multimodal images. Aiming to improve the inaccurate and incomplete results when using a single image segmentation method for image segmentation, Xu et al. [23] applied morphology to Canny edge detection to segment images. The completeness and accuracy of the results were improved after applying the method. Lee et al. [24] proposed a high-speed feature extraction algorithm based on the block type classification algorithm to reduce the computational complexity of feature detection. He applied the proposed method to the Canny edge detection. After experimental verification, it could eliminate the latency of adaptive threshold value calculation. Chen et al. [25] proposed a radar remote sensing image retrieval algorithm that was based on feature vector information. He used an improved Sobel operator to get the edge image. Then, the radar remote sensing image was analyzed from different angles, and the statistic was recorded as a feature-describing vector. Using the feature-describing vector of images can greatly reduce the retrieval time.

The algorithms based on feature points with good stability and strong noise resistance have been widely used in image registration. The feature lines and feature surfaces are easily broken when extracted [26], resulting in incomplete feature lines. It is difficult to achieve a one-to-one correspondence between the two images, and the extraction process is more time-consuming, so it is less used. At this stage, there is no single feature- matching strategy that can solve all registration problems, and different methods should be chosen to register different images by combining their features in different scenarios. In some studies, multi-features are used in the registration process. Xie et al. [27] proposed a region- and feature-based approach for medical image registration in 2018, which is a coarse-to-fine framework. In the coarse registration stage, the closed contours of the target are extracted for a fast initial transformation of the global range; in the fine registration stage, an accurate final transformation is achieved using the modified feature neighborhood and mutual information. Su et al. [28] proposed a method that combined feature points and feature lines for image registration. Firstly, he used LSD to extract the feature lines for coarse registration. Then, the Harris operator was used for fine registration.

In this paper, different feature elements are used for matching in different scenarios. When the map is a remote sensing image or there are relatively more geographical elements in the map, feature points are used for matching; when there are few geographical elements in the map, feature surfaces are used for matching, and a feature surface matching strategy, which combines the geographic feature elements, is used to improve the matching success rate. There is no scale or angle change between tile maps of the same area, and the registration process is a rigid body transformation [29]. The registration process of the method is shown in Figure 1:

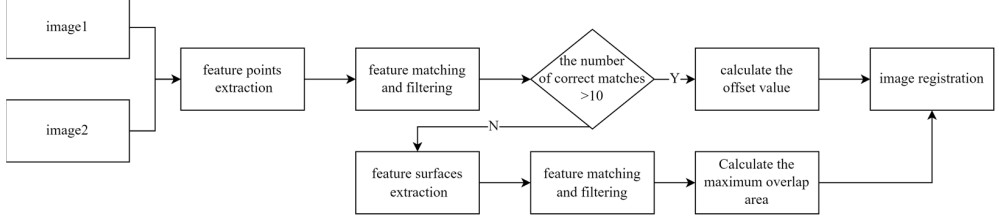

**Figure 1.** Tile map registration flowchart.

## 3. Materials and Methods

### 3.1. SURF Feature Points

SURF extraction is fast with good invariance and robustness and performs well in the feature extraction process [30]. Therefore, we chose the SURF algorithm to extract feature points. The extraction process of the SURF algorithm includes three steps: feature point detection, determining the main direction, and calculating the descriptors.

#### 3.1.1. Feature Points Detection

In order to ensure the scale invariance of SURF feature points, the algorithm performs feature extraction in the same scale space as the SIFT algorithm. The scale space of the image is defined as

$$L(x, y, \sigma) = G(x, y, \sigma) * I(x, y) \tag{1}$$

In the above formula, $\sigma$ represents the scale space, $G(x, y, \sigma)$ is the two-dimensional Gaussian function, and $*$ represents two-dimensional convolution.

To improve the convolution speed, the SURF algorithm uses the box filters (as shown in Figure 2) and uses the integral image to accelerate the image; the calculation efficiency is more than three times higher than that of the SIFT algorithm.

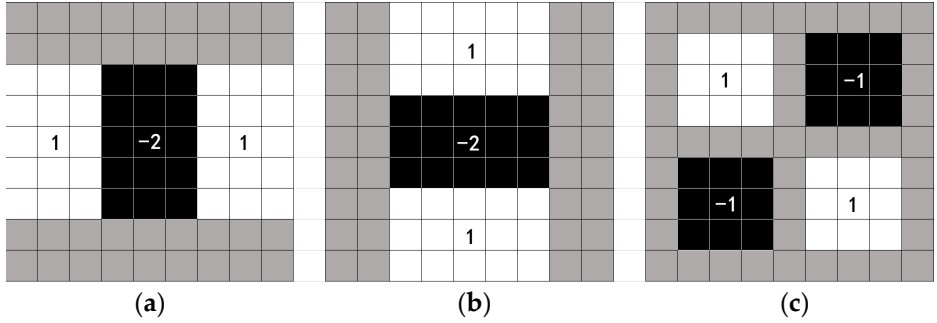

**Figure 2.** Approximations using box filters. (**a**) x direction. (**b**) y direction. (**c**) xy direction.

The extreme values of the pixel points are obtained using the Hessian matrix. The definition of the Hessian matrix at the position $\sigma$ of the image $I(x, y)$ is:

$$H = \begin{bmatrix} L_{xx}(x, \sigma) & L_{xy}(x, \sigma) \\ L_{xy}(x, \sigma) & L_{yy}(x, \sigma) \end{bmatrix} \tag{2}$$

In the above formula, $L_{xx}(x, \sigma)$ is the two-dimensional convolution of the second-order partial derivative of the Gaussian function and the image; $L_{xy}(x, \sigma)$ and $L_{yy}(x, \sigma)$ are defined similarly. Nonmaximal suppression is performed in the stereo neighborhood in the range of $3 \times 3 \times 3$ around the point (as shown in Figure 3), and the candidate feature points must be extreme in the neighborhood range. The candidate feature points are interpolated in the scale space and the image space to obtain the feature point positions and the corresponding scales.

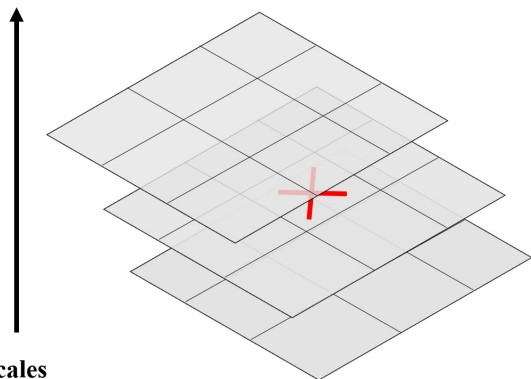

**scales**

**Figure 3.** The $3 \times 3 \times 3$ area where the non-maximum suppression is located.

### 3.1.2. Determine the Main Direction

SURF feature points are invariant to rotation because each feature point has a main direction. The SURF algorithm is used to calculate the Haar wavelet response dx and dy of all pixels in the X and Y directions within a circular range, with the target point as the center and 6σ as the radius, and to assign different weights according to the distance between the pixel point and the center point value, where the closer the distance, the greater the weight. Then, we take a sector with a radius of π/3 as the sliding window, calculate the sum of the responses in the sliding window, and the maximum direction is the main direction of the point. In some cases, there is no rotation variance between two images.

Bay proposed upright SURF(U-SURF) to solve this kind of problem, to which the problem in this paper belongs. Compared with SURF, U-SURF is faster with the same stability.

### 3.1.3. Feature Points Description

To construct the descriptors, a square range with a side length of 20 scales centered around the feature point is determined first. Then, the squares are rotated along the direction of the feature point to ensure the rotational invariance of the feature point. As shown in the figure, the square is partitioned into $4 \times 4$ subregions, each containing 25 sampled pixels. For the 25 points in each subregion, the gradients in the x-direction and y-direction are calculated using Haar wavelets, denoted as dx and dy. The values of the 25 points and the corresponding absolute values are summed to obtain a four-dimensional vector as follows:

$$v = [\sum dx, \sum dy, \sum |dx|, \sum |dy|] \tag{3}$$

The $4 \times 4$ subregions are combined to form a $4 \times 4 \times 4$ feature vector, which is the descriptor of SURF.

### 3.2. Feature Matching Filtering Algorithm for Tile Maps

There are no rotation and scale changes in the tile maps of different manufacturers in the same area, so the expectation of the image coordinate difference of the corresponding points in the two maps is a certain fixed value. The relative relationship between geographical elements of remote sensing maps from different manufacturers is definite, and the pixel difference between the matching point pairs is relatively close. However, the position of some elements in the vector map is inaccurate, such as the position of the text annotation being uncertain, and the edge of the range of the river and the green space is also inaccurate during the measurement process. There are obvious differences between such elements of different manufacturers, which make the differences between pairs of matched points vary greatly.

To calculate the pixel difference between matching pairs of feature points, we use the mean shift algorithm [31] to obtain results. The center value of the clustering point with the highest density is selected as the offset value by filtering from a large number of

matching feature points. The workflow of the algorithm is to start from a certain center point, determine a circle with a radius of D, and continuously offset the center point until the condition is met.

### 3.3. Feature Surfaces Extraction

When the map is zoomed in, the geographical elements of the vector map are reduced, and feature point pairs cannot be obtained stably. Figure 4 shows the results of matching and filtering in the vector map of urban areas. After filtering by rules, 54 pairs of matching points can be obtained. To clearly show the matching result, 11 pairs of the highest rated feature point are selected for display on the map. Figure 5 shows the results of feature point match filtering in nonurban areas. Because the matching feature points are too scattered, the algorithm cannot extract the center of matches, with no matching point pairs obtained.

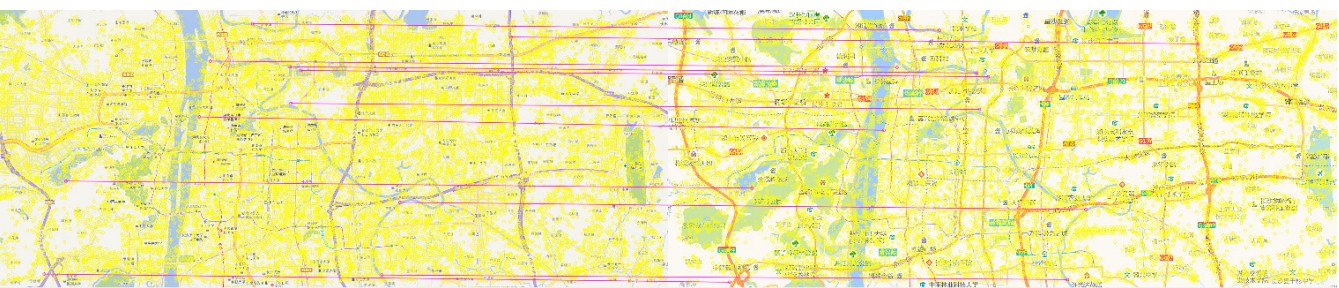

**Figure 4.** Results after matching and filtering in the urban area vector map.

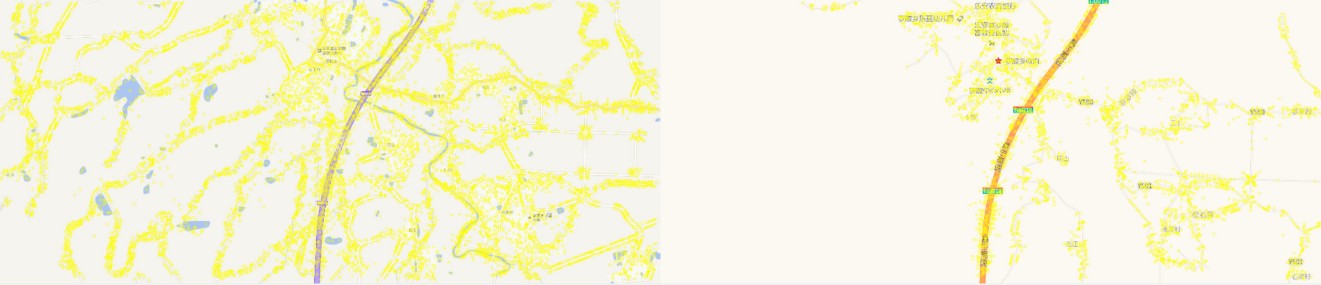

**Figure 5.** Results after matching and filtering in the vector map in nonurban areas.

On a large-scale map, the details of a single element are enlarged, and the geographical elements such as road lines become the main body of the map. These geographical elements are mainly roads, rivers, parks, labels, etc. The main roads are manually measured; their coordinates are more accurate, and they are drawn in the same way in different types of maps. Then, find the corresponding road surface in the two maps, translate the corresponding polygon, and calculate the overlapping area of the two polygons after translation. The offset of the maximum overlapping area is the map offset value.

Suzuki [32] proposed a contour extraction algorithm in 1985, which can be used to extract the road surface. Its implementation process is to scan the image, find the boundary and give each boundary a unique mark, and each boundary also records the mother boundary of the current tracking boundary. The input image is scanned in the order of top to bottom and left to right. The first point that meets the boundary point conditions is marked as the starting point of boundary tracking. Each newly found boundary is given a different numerical label, wherein NBD represents the label of the currently newly found boundary, and LNBD represents the label of the last found boundary. In the process of boundary tracking, the rightmost pixel of the outer boundary is marked as -NBD, which means that the label of this point will never be modified. If the point has not been marked, mark it as NBD, and encounter a point that has been marked. When the last point in the lower right corner of the image is scanned, the algorithm ends.

### 3.4. Feature Surface Matching and Filtering

There are surface breaks in the feature surfaces extraction process, resulting in incomplete extracted surfaces, so the matching process cannot be matched through the characteristics of the extracted surfaces. The coordinates of the feature surfaces between the two images are similar, and whether the surfaces are a pair of matching polygons can be judged by the degree of overlap between the surfaces.

The core of the algorithm is to extract the road lines in the image and filter the matched surfaces from the surface's area, roundness, curvature, and other elements. The filtered results are shown in Figure 6.

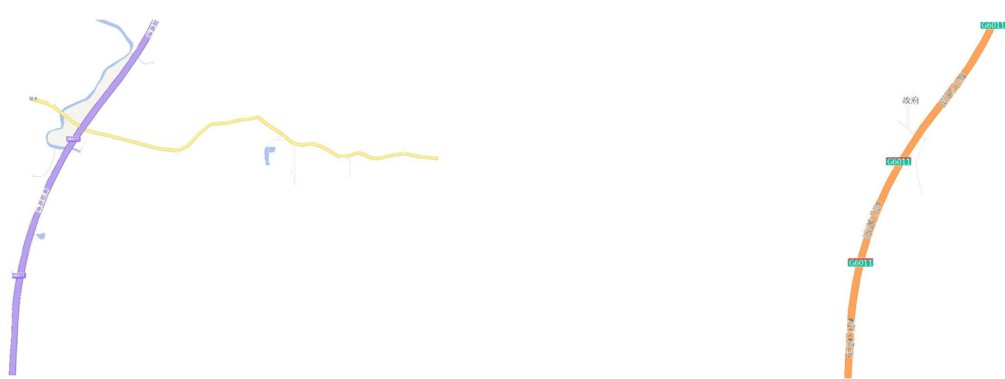

**Figure 6.** Results after feature surfaces matching filtering.

### 3.5. Offset Value Caching and Coarse Registration

To improve the degree of overlap between the two images and reduce the error when the method fails, the offset values of some control points, such as the national core provincial capitals of China, are cached before the method is registered. Before fine registration, the cached value closest to the center of the current map is used to carry out registration, and then the current images are obtained. In addition, the offset values calculated each time are cached in the database to achieve progressive registration during the map zoom-in process.

## 4. Experimental Results and Applications

### 4.1. Method Implementation and Platform Construction

We build a separate front-end and back-end system to implement the method of this paper. The front-end uses VUE + OpenLayers to load the map, and the back-end uses Python + OpenCV to implement the method and encapsulate it into an interface using the Flask framework.

### 4.2. Method Accuracy Verification and Comparison

Taking the Amap as the map to be registered and the Tianditu as the target map, register the different areas of the remote sensing maps and the vector maps, and calculate the root mean square error (RMSE) of the results. RMSE is defined as follows:

$$\text{RMSE} = \sqrt{\frac{\sum_{i=1}^{n}\left(\left(x_i - x_i'\right)^2 + \left(y_i - y_i'\right)^2\right)}{n}} \tag{4}$$

In the above formula, n represents the number of corresponding points, $(x_i, y_i)$ represents the pixel coordinates of the ith corresponding point in the target image, $(x_i', y_i')$ represents the pixel coordinates of the ith corresponding point in the image to be registered.

Map features are important factors affecting the registration accuracy. To test the registration accuracy of PDR in different regions, three different types of regions were selected as the study objects, with Beijing city as a representative of large cities, Ganzhou city as a representative of small and medium-sized cities, and Shaxi Town as a representative

of rural areas. Beijing City is the capital of China, with a regular and complex urban road network. The coordinates of the center of Beijing are [116.39, 39.92]. Ganzhou City is a prefecture-level city in Jiangxi Province. Zhangshui river passes through the center of the city. The features of the image are obvious. The geographical coordinates of the city's center are [114.94, 25.83]. Shaxi Town is located in Yongfeng County, Ji'an City, Jiangxi Province. It belongs to a semi-mountainous and semi-hilly area. Most of the images are forests. The vector map is mainly road lines, and the geographical coordinates are [115.59, 26.88]. In urban areas, the maximum zoom level of remote sensing images of the Amap and Tianditu can reach 18 levels. In nonurban areas, the highest map scale of Amap is level 16, and the highest map scale of Tianditu is up to level 18.

Taking Tianditu scale 12 as an example, Figure 7 shows the remote sensing images and vector map images of the two regions. For the multiscale maps of the two regions, we compare the RMSE of the PDR method, the SURF+RANSAC method, and the coordinate system conversion method. Tables 1–6 show the RMSE and the calculated offset values when using the three methods for registration at multiple scales. "/" means there are no data in this scale, and "×" means the registration method fails in this scale. Table 7 shows the average RMSE of the three methods on remote sensing maps of Beijing City, remote sensing maps of Ganzhou City, remote sensing maps of Shaxi Town, vector maps of Beijing City, vector maps of Ganzhou City, and vector maps of Shaxi Town.

The average RMSE of the PDR method is less than 2.5 in all three areas of remote sensing maps and vector maps, which have good stability and accuracy that can meet the needs of actual use.

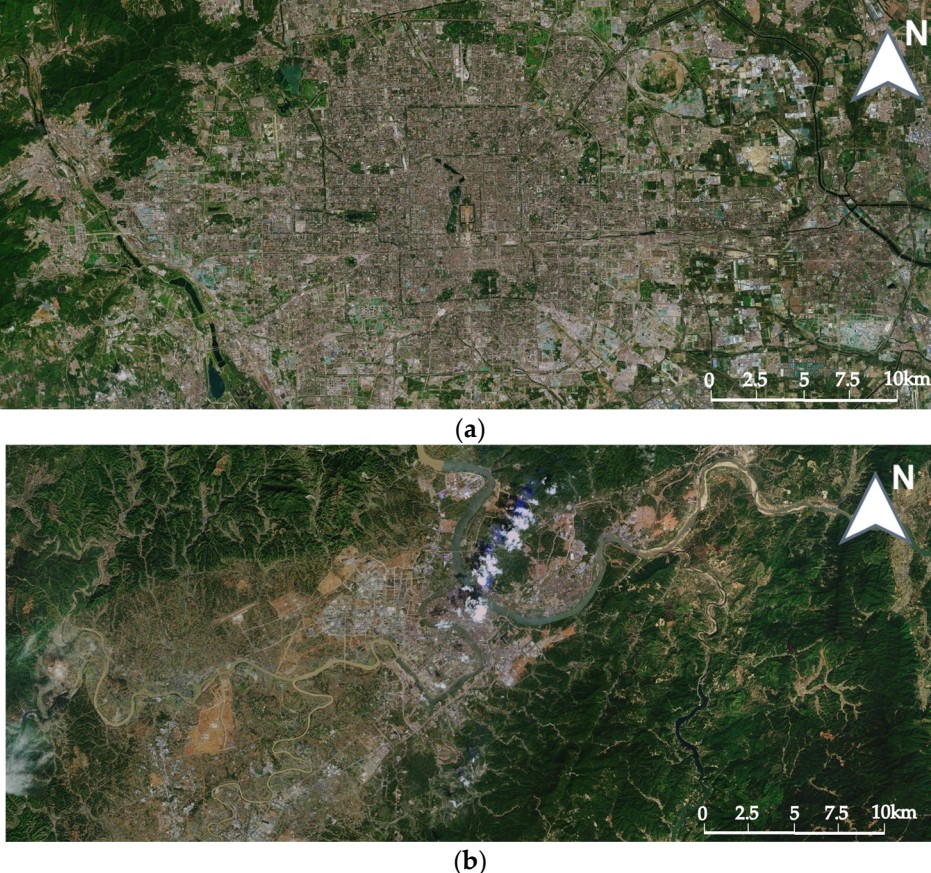

(a)

(b)

**Figure 7.** *Cont.*

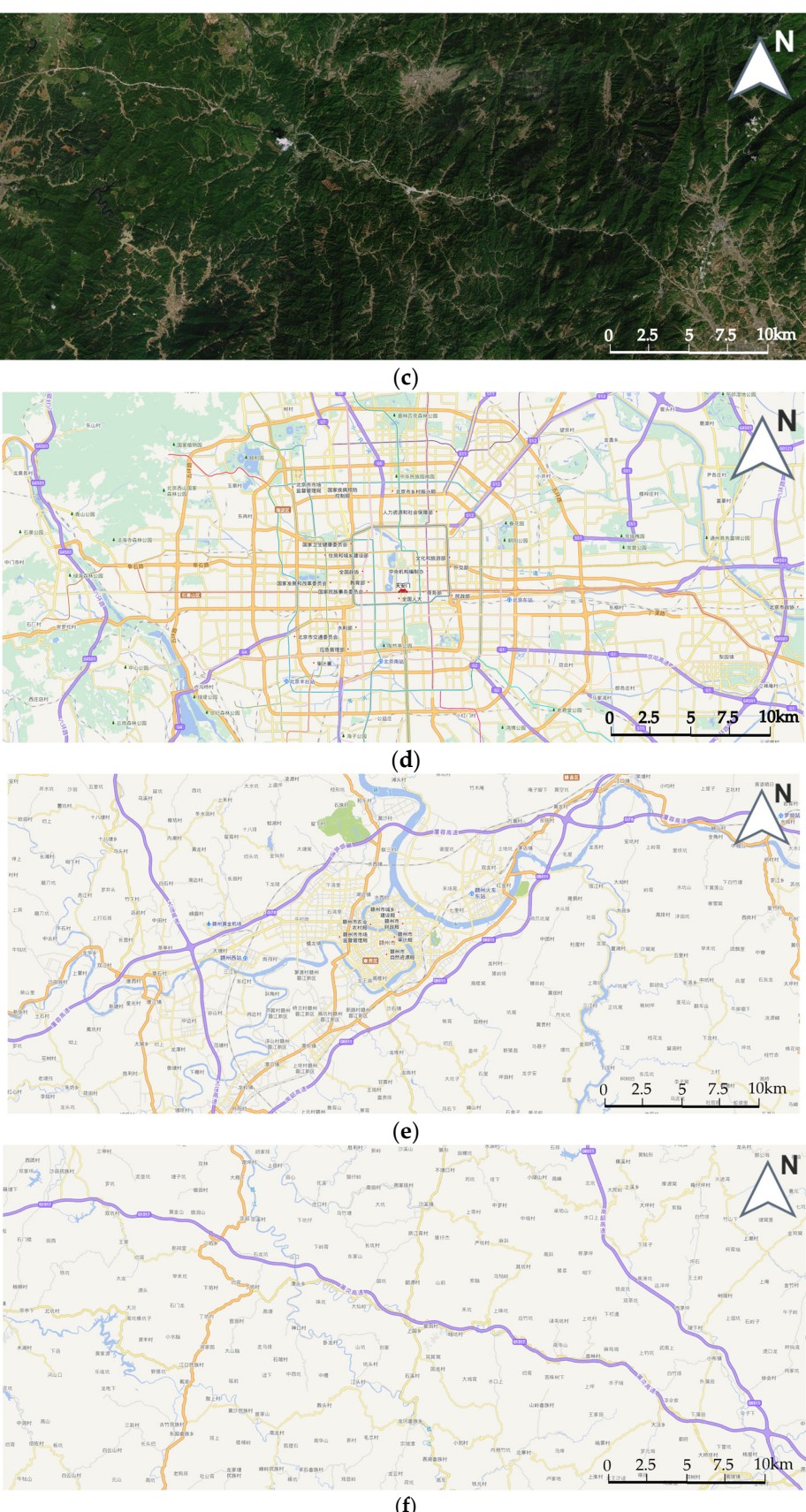

**Figure 7.** Remote sensing maps and vector maps of Tianditu Beijing, Ganzhou and Shaxi. (**a**) Remote sensing map of Beijing. (**b**) Remote sensing map of Ganzhou. (**c**) Remote sensing map of Shaxi. (**d**) Vector map of Beijing. (**e**) Vector map of Ganzhou. (**f**) Vector map of Shaxi.

**Table 1.** Registration results of three methods in remote sensing maps of Beijing.

| ID | Scale | The Method of This Paper | | SURF + RANSAC | | Coordinate System Conversion | |
|----|-------|------|--------------|------|--------------|------|--------------|
| | | RMSE | Offset Value | RMSE | Offset Value | RMSE | Offset Value |
| 1 | 12 | 1.291 | [−680.28, −139.11] | 1.324 | [−676.84, −137.58] | 3.605 | [−694.99, −203.51] |
| 2 | 12.5 | 1.527 | [−684.03, −122.72] | 1.568 | [−688.48, −120.28] | 3.635 | [−694.99, −203.51] |
| 3 | 13 | 1.333 | [−679.31, −118.99] | 1.333 | [−677.45, −118.03] | 5.033 | [−694.99, −203.51] |
| 4 | 13.5 | 1.62 | [−693.16, −202.09] | 4.108 | [−663.88, −229.80] | 1.658 | [−694.99, −203.51] |
| 5 | 14 | 1.287 | [−706.43, −201.95] | 3.553 | [−671.63, −178.48] | 1.5 | [−694.99, −203.51] |
| 6 | 14.5 | 1.458 | [−698.70, −204.64] | 2.937 | [−705.34, −200.59] | 2 | [−694.99, −203.51] |
| 7 | 15 | 1.362 | [−704.97, −204.28] | 1.368 | [−704.00, −203.02] | 3.221 | [−694.99, −203.51] |
| 8 | 15.5 | 1.472 | [−705.70, −206.15] | 1.4 | [−705.77, −202.39] | 3.605 | [−694.99, −203.51] |
| 9 | 16 | 2.449 | [−705.00, −205.67] | 2.543 | [−706.97, −202.14] | 3.949 | [−694.99, −203.51] |
| 10 | 16.5 | 1.581 | [−702.93, −204.26] | 1.87 | [−705.38, −205.12] | 5.597 | [−694.99, −203.51] |
| 11 | 17 | 3.098 | [−705.39, −203.82] | 2.984 | [−704.20, −202.25] | 7.303 | [−694.99, −203.51] |
| 12 | 17.5 | 3.937 | [−705.06, −202.49] | 5.782 | [−706.76, −201.81] | 9.273 | [−694.99, −203.51] |

**Table 2.** Registration results of three methods in remote sensing maps of Ganzhou.

| ID | Scale | The Method of This Paper | | SURF + RANSAC | | Coordinate System Conversion | |
|----|-------|------|--------------|------|--------------|------|--------------|
| | | RMSE | Offset Value | RMSE | Offset Value | RMSE | Offset Value |
| 1 | 12 | 2.683 | [−549.96, 385.62] | 2.652 | [−550.27, 385.70] | 2.489 | [−528.19, 391.93] |
| 2 | 12.5 | 2.669 | [−553.66, 361.19] | 2.703 | [−554.55, 362.37] | 3.316 | [−528.19, 391.93] |
| 3 | 13 | 2.420 | [−536.20, 371.06] | 2.412 | [−537.07, 372.18] | 3.696 | [−528.19, 391.93] |
| 4 | 13.5 | 1.699 | [−535.93, 381.82] | 1.682 | [−536.55, 382.91] | 1.943 | [−528.19, 391.93] |
| 5 | 14 | 1.974 | [−524.84, 383.79] | 1.874 | [−524.32, 384.86] | 1.923 | [−528.19, 391.93] |
| 6 | 14.5 | 2.000 | [−538.46, 381.90] | 1.968 | [−535.77, 382.71] | 1.871 | [−528.19, 391.93] |
| 7 | 15 | 1.871 | [−529.39, 391.25] | 2.013 | [−530.75, 390.17] | 1.897 | [−528.19, 391.93] |
| 8 | 15.5 | 2.683 | [−527.32, 395.63] | 2.624 | [−527.21, 395.44] | 2.387 | [−528.19, 391.93] |
| 9 | 16 | 2.366 | [−529.00, 390.39] | 2.311 | [−530.73, 391.31] | 2.549 | [−528.19, 391.93] |
| 10 | 16.5 | 2.366 | [−535.42, 390.18] | 3.412 | [−535.23, 390.49] | 4.393 | [−528.19, 391.93] |
| 11 | 17 | 2.185 | [−533.68, 390.44] | 6.944 | [−528.55, 385.21] | 5.132 | [−528.19, 391.93] |
| 12 | 17.5 | 5.060 | [−537.56, 387.36] | 14.352 | [−542.44, 381.11] | 8.615 | [−528.19, 391.93] |

**Table 3.** Registration results of three methods in remote sensing maps of Shaxi.

| ID | Scale | PDR | | SURF + RANSAC | | Coordinate System Conversion | |
|----|-------|------|--------------|------|--------------|------|--------------|
| | | RMSE | Offset Value | RMSE | Offset Value | RMSE | Offset Value |
| 1 | 12 | 2.104 | [−572.66, 408.70] | 2.138 | [−572.51, 408.55] | 2.672 | [−539.63, 442.98] |
| 2 | 12.5 | 1.288 | [−568.57, 401.85] | 1.309 | [−578.05, 411.19] | 2.408 | [−539.63, 442.98] |
| 3 | 13 | 1.414 | [−568.88, 401.50] | 2.273 | [−560.02, 396.42] | 3.400 | [−539.63,442.98] |
| 4 | 13.5 | 1.673 | [−548.88, 427.80] | 2.236 | [−546.42, 439.11] | 1.703 | [−539.63, 442.98] |
| 5 | 14 | 1.511 | [−535.88, 433.33] | 1.690 | [−536.74, 445.55] | 1.690 | [−539.63, 442.98] |
| 6 | 14.5 | 1.510 | [−535.08, 441.58] | 1.541 | [−532.93, 444.42] | 1.370 | [−539.63, 442.98] |
| 7 | 15 | 1.824 | [−538.32, 442.23] | 2.170 | [−533.42, 442.93] | 1.050 | [−539.63, 442.98] |
| 8 | 15.5 | 2.160 | [−538.70, 442.10] | 2.366 | [−530.70, 441.67] | 1.370 | [−539.63, 442.98] |

**Table 4.** Registration results of three methods in vector maps of Beijing.

| ID | Scale | The Method of This Paper | | SURF + RANSAC | | Coordinate System Conversion | |
|---|---|---|---|---|---|---|---|
| | | RMSE | Offset Value | RMSE | Offset Value | RMSE | Offset Value |
| 1 | 12 | 1.224 | [−639.01, −179.24] | 6.066 | [−845.39, −33.63] | 1.541 | [−694.99, −203.51] |
| 2 | 12.5 | 1.527 | [−681.77, −154.08] | 586 | [14,892.15, −5420.31] | 1.695 | [−694.99, −203.51] |
| 3 | 13 | 1.526 | [−688.78, −165.73] | × | × | 1.105 | [−694.99, −203.51] |
| 4 | 13.5 | 1.105 | [−679.20, −198.53] | × | × | 1.732 | [−694.99, −203.51] |
| 5 | 14 | 1.699 | [−684.53, −185.83] | × | × | 1.452 | [−694.99, −203.51] |
| 6 | 14.5 | 1.452 | [−692.79, −199.13] | × | × | 1.563 | [−694.99, −203.51] |
| 7 | 15 | 1.291 | [−698.72, −208.52] | × | × | 1.414 | [−694.99, −203.51] |
| 8 | 15.5 | 2.108 | [−698.72, −208.52] | × | × | 1.943 | [−694.99, −203.51] |
| 9 | 16 | 2.397 | [−699.74, −205.39] | × | × | 2.091 | [−694.99, −203.51] |
| 10 | 16.5 | 2.298 | [−696.56, −206.49] | × | × | 2.329 | [−694.99, −203.51] |

**Table 5.** Registration results of three methods in vector maps of Ganzhou.

| ID | Scale | PDR | | SURF + RANSAC | | Coordinate System Conversion | |
|---|---|---|---|---|---|---|---|
| | | RMSE | Offset Value | RMSE | Offset Value | RMSE | Offset Value |
| 1 | 12 | 1.768 | [−610.08, 414.25] | 1.760 | [−618.52, 408.70] | 1.732 | [−528.19, 391.93] |
| 2 | 12.5 | 2.031 | [−534.30, 424.22] | 8.062 | [−732.92, 392.56] | 1.369 | [−528.19, 391.93] |
| 3 | 13 | 1.581 | [−515.19, 405.11] | 145.710 | [1226.34, 2516.97] | 1.767 | [−528.19, 391.93] |
| 4 | 13.5 | 2.168 | [−555.72, 364.57] | 605.070 | [8619.72, 4837.16] | 1.581 | [−528.19, 391.93] |
| 5 | 14 | 3.000 | [−546.17, 374.12] | 1116.360 | [9711.62, 3414.85] | 2.055 | [−528.19, 391.93] |
| 6 | 14.5 | 2.236 | [−539.41, 380.88] | × | × | 1.363 | [−528.19, 391.93] |
| 7 | 15 | 3.644 | [−544.19, 380.88] | × | × | 2.645 | [−528.19, 391.93] |
| 8 | 15.5 | 2.203 | [−523.92, 391.01] | × | × | 1.963 | [−528.19, 391.93] |
| 9 | 16 | 1.603 | [−528.69, 393.40] | × | × | 2.070 | [−528.19, 391.93] |
| 10 | 16.5 | 3.768 | [−525.31, 386.64] | × | × | 3.130 | [−528.19, 391.93] |

**Table 6.** Registration results of three methods in vector maps of Shaxi.

| ID | Scale | PDR | | SURF + RANSAC | | Coordinate System Conversion | |
|---|---|---|---|---|---|---|---|
| | | RMSE | Offset Value | RMSE | Offset Value | RMSE | Offset Value |
| 1 | 12 | 1.580 | [−581.30, 400.18] | 3.460 | [−650.86, 429.19] | 1.603 | [−539.63, 442.98] |
| 2 | 12.5 | 2.000 | [−527.25, 454.23] | 7.000 | [−764.14, 464.32] | 1.914 | [−539.63, 442.98] |
| 3 | 13 | 1.825 | [−565.46, 396.90] | 7.071 | [−435.08, 357.04] | 2.000 | [−539.63, 442.98] |
| 4 | 13.5 | 1.224 | [−551.95, 410.41] | 8.940 | [−429.81, 405.41] | 2.041 | [−539.63, 442.98] |
| 5 | 14 | 1.802 | [−561.50, 448.63] | 5.000 | [−538.63, 401.97] | 2.031 | [−539.63, 442.98] |
| 6 | 14.5 | 2.041 | [−541.24, 455.38] | 10.198 | [−474.47, 446.21] | 1.914 | [−539.63, 442.98] |
| 7 | 15 | 1.612 | [−541.24, 436.27] | 161.500 | [−1297.60, 351.95] | 2.049 | [−539.63, 442.98] |
| 8 | 15.5 | 2.738 | [−551.37, 453.16] | 485.060 | [−2139.85, 802.55] | 1.732 | [−539.63, 442.98] |
| 9 | 16 | 1.414 | [−538.85, 447.02] | 918.190 | [−2657.80, 1029.11] | 2.291 | [−539.63, 442.98] |
| 10 | 16.5 | 1.264 | [−538.85, 445.33] | 1500.000 | [−2657.80, 1029.11] | 1.732 | [−539.63, 442.98] |

**Table 7.** RMSE of registration for three methods in six scenarios.

| Type | Method | RMSE | | |
|---|---|---|---|---|
| | | Beijing | Ganzhou | Shaxi |
| Remote sensing map | PDR | 1.86 | 2.49 | 1.68 |
| | SURF + RANSAC | 2.56 | 3.52 | 1.96 |
| | coordinate system conversion | 4.19 | 3.35 | 1.95 |
| Vector map | PDR | 1.66 | 2.4 | 1.75 |
| | SURF + RANSAC | 586 | 1116.36 | 310.64 |
| | coordinate system conversion | 1.68 | 1.96 | 1.93 |

### 4.3. Vector Data Registration

Some maps come with vector data services, and the coordinate system of the vector data is often the same as the map. For example, users can download the nodes, road lines, and area surfaces of a specified area from the official website of OpenStreetMap. The method of this paper is used to obtain the target area offset values and perform dynamic registration of the vector data.

Figure 8a shows the effect of using Open Street Map as the base map and overlaying the downloaded vector road network of the Central South University (CSU), and Figure 8b shows the effect of using Amap as the base map and overlaying this vector road network, with an obvious offset.

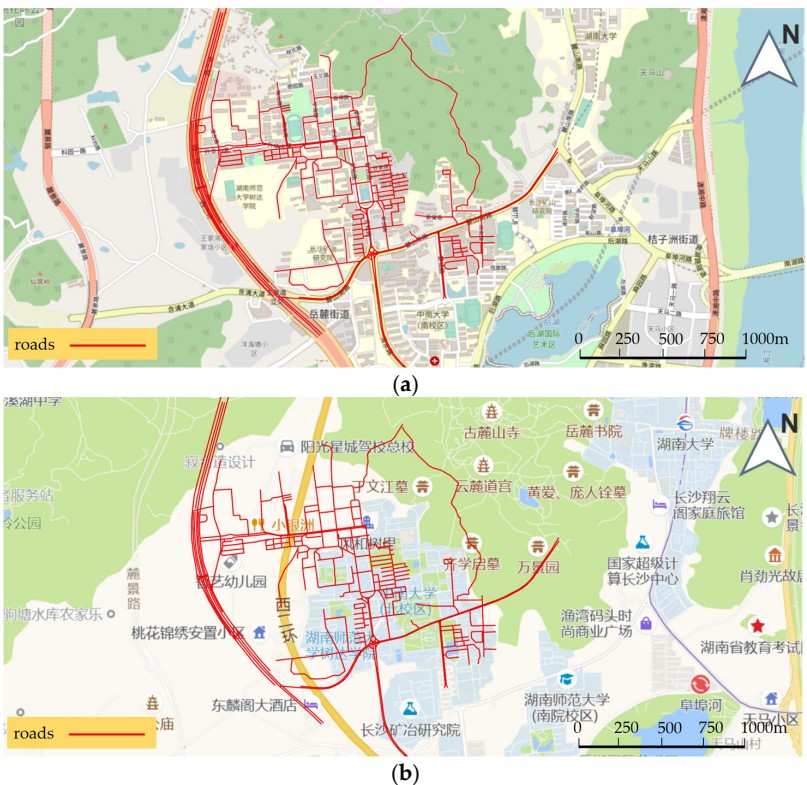

**Figure 8.** Vector data of CSU with different base map. (**a**) Vector data of CSU with Open Street Map as the base map. (**b**) Vector data of CSU with Amap as the base map.

The offset value between Open Street Map and Amap is calculated using the method of this paper to register the vector data, and the calculation result is [−593.90, 474.35]. The result of the registration is shown in Figure 9.

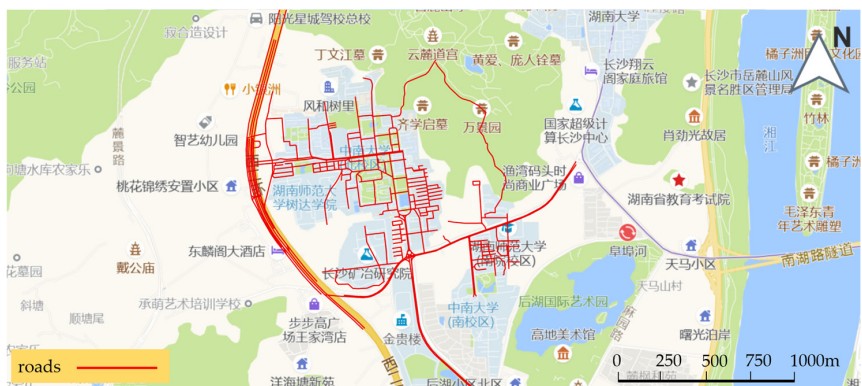

**Figure 9.** Registration result of the vector data.

## 5. Discussion

In the six scenarios (remote sensing maps of big city, remote sensing maps of small city, remote sensing maps of nonurban areas, vector maps of big city, vector maps of small city, and vector maps of nonurban areas), the average RMSE of the PDR method is lower than that of the SURF + RANSAC method, especially in vector map. Furthermore, the PDR method is more stable. This is because the image difference between the two vector maps is obvious, and it is difficult to match enough corresponding point pairs according to the descriptor information of the feature points. On the one hand, we propose a feature point filtering strategy for the characteristics of tile maps to improve the match success rate. On the other hand, when a certain proportion of correct matching feature point pairs cannot be obtained, feature surfaces are used for registration. Therefore, the PDR method performs better.

Compared with the method using coordinate system conversion, the registration effect of the PDR method is better in remote sensing maps and vector maps in nonurban areas and is poorer in vector maps of some urban areas. The remote sensing maps of different manufacturers are different images of the same features, and there are many corresponding points between images. Using the PDR method for registration, a large number of correct matching point pairs can be obtained stably, and the calculation results are accurate. In contrast, the remote sensing maps of map manufacturers are made from many remote sensing images stitched and encrypted in bulk using various methods, and the accuracy is low. Therefore, better registration results can be achieved in both urban and nonurban areas. However, the drawing style of vector maps varies greatly among different manufacturers, and the measurement coordinates of the same element may be different. It is difficult for the registration method based on image features to achieve stable registration results. In the vector map of urban areas, when the map scale is small, urban arterials with accurate coordinates are the main part of the map, and there are fewer text and road signs with inaccurate coordinates in the map, so stable feature point pairs can be extracted, and the RMSE of the PDR method is low. As the scale increases, there are fewer urban arterials and more minor roads and boundaries of the community, which is unreliable; it is difficult to obtain enough feature point pairs after using the algorithm filtering, and then feature surfaces are used for registration. The inaccurate polygons of minor roads and community boundaries in urban areas make the extraction of polygons difficult, and the RMSE is high. In large cities, the road system is more developed, and the coordinates are more accurate, so more feature points are extracted, fewer scenes are registered using feature surfaces, and the RMSE of the PDR method is lower than in small cities. In the experiments, feature surfaces are used for registration from zoom 16.5 in Beijing, while it is from zoom 14 in Ganzhou. In the vector map of non-urban areas, there are fewer geographic elements, and it is easy to extract the corresponding geographic feature surfaces, and the registration effect is better.

## 6. Conclusions

In this paper, a progressive dynamic registration (PDR) method is proposed for tile maps. With the PDR method, we can perform registration between any two tile maps of the same type without knowing the coordinate system conversion formula between the maps. Moreover, the PDR method has strong versatility; it can not only be used to register the tile maps but also the corresponding vector data. When there are many image feature elements, we use feature points for matching and calculation, and when there are few image feature elements, we use feature surfaces for matching and calculation. We combine the characteristics of the tile map and propose a matching filtering strategy, which improves the correct matching rate. The values of adjacent levels are used as references in the registration process to progressively optimize the results. The root mean square error of the PDR method is below 2.5 in different maps. Compared with registration using feature points only, the accuracy of the PDR method is significantly improved; compared with the method using coordinate system conversion, the registration effect of the PDR method is

better in remote sensing maps and vector maps in nonurban areas, while it needs to be improved in vector maps of urban areas.

**Author Contributions:** Conceptualization, D.Z. and J.D.; data curation, D.Z.; formal analysis, J.D.; funding acquisition, J.D.; investigation, D.Z.; methodology, D.Z. and J.D.; project administration, D.Z.; software, D.Z.; supervision, J.D.; validation, D.Z.; visualization, D.Z.; writing—original draft, D.Z.; writing—review and editing, J.D. All authors have read and agreed to the published version of the manuscript.

**Funding:** National Natural Science Foundation of China (Grant Number: 42172330).

**Institutional Review Board Statement:** Not applicable.

**Informed Consent Statement:** Not applicable.

**Data Availability Statement:** Not applicable.

**Conflicts of Interest:** The authors declare no conflict of interest.

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
