# Peer review of "Progressive Dynamic Registration Method for Tile Maps Based on Optimum Multi-Features"

_applsci, doi:10.3390/app13074282_

Round 1

Reviewer 1 Report

This study proposes a progressive dynamic registration method based on the best features extracted from a certain scale of tile map screenshot images. 

The authors fully present the current state of the registration method for tile maps in the introduction. 

However, I still have the following questions and expect responses from the authors.

(1) The authors chose only Ganzhou and Shaxi as the experimental areas. Do these two study areas adequately illustrate the validity of the method? From another perspective, why did you choose these two areas? Are they somehow representative? I suggest that the authors could add experimental areas of different scales for comparison. For example, mega-city, large city, medium city, etc. Since there are large differences in the map elements of various scales of cities.

(2)Does the PDR method work for remote sensing image matching? As far as I know, the elements of remote sensing images are more complex and you may detect more feature points and there will be more mismatches.

Finally, I have made the following comments and suggestions for the author to further improve.

(1) I found quite a few spelling errors in the article, such as line 273, where ‘python’ is missing a ‘y’.

(2) Does the ‘Gaode map’ is ‘Amap’? And what is Zhangshui? A road or a river? Open Street Map in line.324. The author should please further confirm the professional translation of these applications and terminologies. 

(3) Please include an explanation, e.g. CSU, when using abbreviations. Is this the author's institution/university?

(4) Your maps are missing essential map elements, such as scale and compass. This does not look very professional. I also suggest that authors take more time for proofreading the language.

Author Response

Thank you very much for your comments on our paper! Your suggestions are very important for our research.

Point 1: The authors chose only Ganzhou and Shaxi as the experimental areas. Do these two study areas adequately illustrate the validity of the method? From another perspective, why did you choose these two areas? Are they somehow representative? I suggest that the authors could add experimental areas of different scales for comparison. For example, mega-city, large city, medium city, etc. Since there are large differences in the map elements of various scales of cities.

Response 1: In the previous experiment, we did not choose a large city, but randomly selected two unrepresentative areas for the experiment to ensure the universality of the experiment. Your suggestion is very informative, and we added the Beijing City to the experiment as a representative of large cities. Now, our experimental areas include three area with high, medium and low anthropogenic elements. In addition, map elements vary with map scales.

Point 2: Does the PDR method work for remote sensing image matching? As far as I know, the elements of remote sensing images are more complex and you may detect more feature points and there will be more mismatches.

Response 2: In all three sets of our experimental data, the registration accuracy of remote sensing images is better than the other two methods. Although the remote sensing images are more complex, the rich features are more favorable for the PDR algorithm registration because the error matches can be screened out to a great extent by our proposed filtering algorithm. The traditional feature point matching method is unable to screen out a large number of false matches, and the complex images will lead to lower accuracy (as shown in Tables 1-6).

Point 3: (1) I found quite a few spelling errors in the article, such as line 273, where ‘python’ is missing a ‘y’.

(2) Does the ‘Gaode map’ is ‘Amap’? And what is Zhangshui? A road or a river? Open Street Map in line.324. The author should please further confirm the professional translation of these applications and terminologies.

(3) Please include an explanation, e.g. CSU, when using abbreviations. Is this the author's institution/university?

(4) Your maps are missing essential map elements, such as scale and compass. This does not look very professional. I also suggest that authors take more time for proofreading the language.

Response 3: I changed all the errors you mentioned, they correspond to (1)=>line 294, (2)=>line 33,line 311,line 359, (3)=>line 363, (4)=>figure 7, figure 8, figure 9. In addition, English editing service recommended by MDPI are undergone.

Reviewer 2 Report

This paper shows progressive dynamic registration  method for tile maps. With this method authors showed how to perform registration between any two tile maps of the same type without knowing the coordinate system conversion formula between maps.  The soulution is  very significant and can be used in practice

Abstract should be more concise and to point out the result of the paper. Also related work should be singled out as a new chapter.

English language should be corrected. The article should be checked for typos.

Figure 4 is unclear. You could show images with less point pairs, then it will be understandable.

Author Response

Thank you very much for your comments on our paper! Your suggestions are very important for our research.

Point 1: Abstract should be more concise and to point out the result of the paper. Also related work should be singled out as a new chapter.

Response 1: We have streamlined the abstract and the current status of the study is placed in a separate section.

Point 2: English language should be corrected. The article should be checked for typos.

Response 2: English editing service recommended by MDPI are undergone. We have made uniform corrections to proper nouns.

Point 3: Figure 4 is unclear. You could show images with less point pairs, then it will be understandable.

Response 3: For easy viewing, we only show the 11 pairs of feature points with the highest ratings. Now, the correspondence of the feature points can be clearly seen in the figure.(line 247)
